# fPLSA: Learning Semantic Structures in Document Collections Using Foundation Models

## Abstract

Humans have the ability to learn new tasks by inferring high-level concepts from existing solutions, then manipulating these concepts in lieu of the raw data. Can we automate this process by deriving latent semantic structures in a document collection using foundation models? We introduce *fPLSA*, a foundation-model-based Probabilistic Latent Semantic Analysis (PLSA) method that iteratively clusters and tags document segments based on document-level contexts. These tags can be used to model the structure of given documents and for hierarchical sampling of new texts. Our experiments on story writing, math, and multi-step reasoning datasets demonstrate that *fPLSA* tags help reconstruct the original texts better than existing tagging methods. Moreover, when used for hierarchical sampling, *fPLSA* produces more diverse outputs with a higher likelihood of hitting the correct answer than direct sampling and hierarchical sampling with existing tagging methods.

## 1 Introduction

Large language models (LLMs) have shown impressive performance on a wide range of tasks, such as reasoning (Suzgun et al., 2022; Liu et al., 2023), math problem solving (Wu et al., 2023), and open-ended text generation tasks (Katz et al., 2024; Dubey et al., 2024; OpenAI et al., 2024). Given natural language instructions or in-context examples with chain-of-thought steps, LLMs can adapt quickly to a new task and achieve outstanding performance on challenging tasks that require multi-step reasoning or planning (Wei et al., 2022). However, such methods typically rely on humans to provide the LLM with instructions or chain-of-thought recipes for solving a task. By contrast, humans can directly derive effective methodologies for solving a task by analyzing a separate set of problems and their solutions.

Can we automate the process of discovering latent semantic structures in a document collection using LLMs? Such algorithms would have a wide range of applications, including producing effective guidelines for new tasks, hierarchical sampling for diverse outputs, and document analysis. For example, they can help determine how two document collections differ in text structure and identify the most common plot elements in a story collection.

We frame this problem as an unsupervised clustering and tagging problem, where we discover the text segments that share common characteristics and assign them to the same tag. Based on these segment tags, we can model the latent structure of a collection of documents by learning a dynamic model over the latent tags and their transitions in the documents. Traditional document labeling and topic modeling approaches focus primarily on lexical features such as word or term co-occurrence (Hearst, 1997; Blei et al., 2003; Hofmann et al., 1999), which provide minimal information on the semantics of short text spans. Recent LLM-based approaches discover topics based on higher-level semantic contexts, but rely on one-shot topic generation and merging (Pham et al., 2024; Wang et al., 2023; Mu et al., 2024), which limits the model's ability to uncover shared characteristics among seemingly unrelated text spans.

In this paper, we introduce *fPLSA*, an iterative algorithm that alternatively clusters and tags document segments using LLMs based on segment- and document-level contexts. *fPLSA* combines the merits of traditional topic modeling approaches such as Probabilistic Latent Semantic Analysis (PLSA) (Hofmann et al., 1999) and LLM-based approaches, and captures shared semantic features among text segments more effectively.

We evaluate the informativeness of *fPLSA* tags by measuring 1) how well they help reconstruct the original text spans, and 2) how useful they are in hierarchical sampling to produce structurally diverse outputs that cover more solution paths. Experiments on story writing, math and multi-step reasoning datasets show that *fPLSA* leads to higher reconstruction likelihood than existing tagging approaches. Furthermore, on math and reasoning tasks, hierarchical sampling using *fPLSA* tags produces more diverse outputs, which increase the probability of hitting the correct answer over hierarchical sampling with other tagging approaches.

## 2    RELATED WORK

### 2.1    DOCUMENT SEGMENTATION AND LABELING

To model the structure and topic shifts in a document, prior work has introduced unsupervised document segmentation and labeling approaches that leverage term co-occurrence features (Hearst, 1997), co-occurrence shifts in topic vectors (Riedl & Biemann, 2012), lexical features and word embeddings (Glavaš et al., 2016). These approaches focus mostly on lexical features which are limited in modeling the high-level semantic structure of documents. On the other hand, Neural-based approaches have the potential of modeling sentence-level semantics and document-level topic flows more effective, but rely heavily on supervised training samples in the target domain (Koshorek et al., 2018; Arnold et al., 2019; Zhang et al., 2019). Our algorithm infers the structure of documents based on segment- and document-level contexts using LLMs in an unsupervised fashion.

### 2.2    TOPIC MODELING

Topic modeling is a widely used technique in natural language processing for uncovering hidden thematic structures in large text corpora. The most foundational methods in this domain include Latent Dirichlet Allocation (LDA) (Blei et al., 2003) and Probabilistic Latent Semantic Analysis (PLSA) (Hofmann et al., 1999; Hofmann, 1999; 2001). Both methods represent each document as a bag of words and models word-document relationships using a mixture of latent topics, where each topic is represented by a list of top words. These algorithms are mathematically grounded, but typically rely on manual topic interpretation, which often leads to incorrect or incomplete labels (Gillings & Hardie, 2022). More recent work introduces neural topic models (Miao et al., 2016; Dieng et al., 2020; Srivastava & Sutton, 2017), which combine traditional topic models with word embeddings. These models have shown improved performance in handling large and complex vocabularies. However, they sill model each document as a bag of words, disregarding the sentence- and document-level semantics. Additionally, the resulting topics are represented either by semantic vectors or lists of closest words, which still rely on manual interpretation. Furthermore, studies have shown that incorporating expert knowledge in topic modeling improves over traditional unsupervised methods (Lee et al., 2017).

Moreover, the advent of large language models (LLMs) has led to LLM-based topic modeling approaches. Li et al. (2023) propose to use LLMs for topic labeling based their top terms produced by traditional topic models. For short text spans, however, the bag-of-words representation of texts provides limited information for topic modeling. Akash et al. (2023) address the issue by extending each text span into longer sequences using LLMs and extracting topics from the extended texts using neural topic models. Futhermore, Pham et al. (2024); Wang et al. (2023); Mu et al. (2024) propose prompt-based techniques to generate, merge, and assign topics using LLMs. These approaches leverage the domain knowledge embedded in LLMs and produce more interpretable topics based on sentence or document-level contexts beyond bag of words.

However, the generate-and-merge approach limits the model's potential for discovering shared features among various text spans across documents of different themes and often leads to overly abstract, thematical topics, especially on a large-scale document collection. We propose *fPLSA*, which combines the merits of traditional PLSA, which uses an iterative EM algorithm to model topic and text distributions, and LLM-based approaches.

## 3 APPROACH

We propose fPLSA, a foundation-model-based EM algorithm that learns the latent tags on a set of segmented documents. We draw inspiration from the traditional Probabilistic Latent Semantic Analysis and use iterative EM steps to learn the latent topics that maximize the estimated likelihood of segmented documents.

### 3.1 PROBABILISTIC LATENT SEMANTIC ANALYSIS (PLSA)

PLSA models the distribution over words $w$ in a document $d$ as a mixture of conditionally independent multinomial distributions, each such distribution representing a *topic* $t$. More formally, the generative model of words in a document can be written as:

$$p_\Theta(w, d) = p(d) \sum_t p_\Theta(t|d) p_\Theta(w|t) \tag{1}$$

where the topic $t$ can be viewed as a discrete latent variable and the total number of discrete topics is pre-defined. $\Theta$ represents the parameters of the PLSA model.

To estimate the parametric distributions $p_\Theta(t|d)$ and $p_\Theta(w|t)$, PLSA relies on an EM algorithm, which is an iterative method to find the maximum likelihood estimate of parameters in statistical models. Specifically, an EM iteration alternates between an expectation (E) step and a maximization (M) step. At iteration $i$, the E-step estimates the posterior distribution of topics $t$ conditioned on each document $d$ and word $w$ in it based on fixed parameters $\Theta_{i-1}$ from the previous iteration:

$$p_{\Theta_{i-1}}(t|w, d) = \frac{p_{\Theta_{i-1}}(t|d) p_{\Theta_{i-1}}(w|t)}{\sum_{t'} p_{\Theta_{i-1}}(t'|d) p_{\Theta_{i-1}}(w|t')} \tag{2}$$

The M-step optimizes the parameters $\Theta$ such that the expectation of the joint distribution $p_\Theta(w, d)$ with $t$ sampled from the estimated posterior $p_{\Theta_{i-1}}(t|w, d)$ is maximized:

$$\Theta_i = \arg\max_\Theta \mathbb{E}_{t \sim p_{\Theta_{i-1}}(t|w,d)} p(d) p_\Theta(t|d) p_\Theta(w|t) \tag{3}$$

Theoretically, each EM iteration will yield a larger likelihood in Eq 1 until it converges to a local maximum.

### 3.2 FOUNDATION-MODEL-BASED PLSA (FPLSA)

We introduce fPLSA, which learns the latent *tags* (similar to *topics* in LSA)[1] on a set of segmented documents $d = (x_1, x_2, ..., x_L)$, where the document $d$ is segmented into $L$ segments $x_k$. A core difference between *fPLSA* and PLSA is that *fPLSA* models the probability of the sequence of words $(w_1, w_2, ..., w_n)$ in each text segment $x_k$ jointly as $p_\Theta(w_1, w_2, ..., w_n|t)$. Moreover, *fPLSA* models the distribution over tags $t$ conditioned not only on current segment $x_k$ but also on the document $d$. Formally, in *fPLSA*, the generative model of a segment $x_k = w_{1...n}$ in a document $d$ can be written as:

$$p_\Theta(w_{1...n}, x_k, d) = p(d) p(x_k|d) \sum_t p_\Theta(t|x_k, d) p_\Theta(w_{1...n}|t) \tag{4}$$

Another core difference between *fPLSA* and PLSA is that we model the parametric distributions $p_\Theta(t|x_k, d)$ and $p_\Theta(w_{1...n}|t)$ using an LLM. Specifically, the parameters $\Theta$ in *fPLSA* include the LLM parameters, which is frozen, and the textual description $\theta_t$ for each tag $t$.

Inspired by PLSA, we also maximize the likelihood in Eq 4 using iterative EM steps.

At the E-step in iteration $i$, we approximate the posterior distribution $p_{\Theta_{i-1}}(t|w_{1...n}, x_k, d)$ of tags $t$ conditioned on each document $d$ and segment $x_k = w_{1...n}$ in it by prompting the LLM to greedily assign a tag given the tag descriptions $\theta_{i-1}$ from the previous iteration, the current segment $x_k = w_{1...n}$ and neighbouring segments $(x_{k-W/2}, x_{k+1-W/2}, ..., x_{k-1+W/2}, x_{k+W/2})$ as document-level context, where $W$ is the context window size.

---

[1]We use the terminology *tag* instead of *topic* in our algorithm because they may cover shared characteristics among document segments beyond topics (see the example tags in Section 5 for more details).

At the M-step, we optimize the tag description $\theta_t$ for each tag $t$ by aggregating the segments assigned to tag $t$ and prompting the LLM to generate a tag description that best summarizes what these segments share in common.

# 4 EXPERIMENTAL SETUP

## 4.1 EVALUATION DATASETS

We evaluate *fPLSA* against various baselines on story writing, math problem solving and multi-step reasoning benchmarks. We use WritingPrompts (Fan et al., 2018), a story writing dataset that contains 300K human-written stories paired with writing prompts from an online forum. We randomly sample 100 stories from the training set for clustering and tagging. We set the number of tags to 100 for all tagging approaches. For math problem solving, we use MATH (Hendrycks et al., 2021), a popular math benchmark that contains high school math competition problems on seven subjects including Prealgebra, Algebra, Number Theory, Counting and Probability, Geometry, Intermediate Algebra and Precalculus. We learn 100 tags on 1K randomly sampled problems and the step-by-step solutions from the training set. We also experiment on the Big-Bench Hard (BBH) benchmark (Suzgun et al., 2022). The original benchmark includes 23 challenging multi-step reasoning tasks, but each task only includes three step-by-step solution examples. Instead, we pick the 12 tasks used in Xu et al. (2024) and use the step-by-step solutions produced by their automatic Chain-of-Thought prompt inference algorithm for clustering and tagging. We set the number of tags to 50 on BBH.

## 4.2 EVALUATION METRICS

We evaluate our approach on two different evaluation protocols.

**Reconstruction Likelihood** To test how well the learned tags help predict the original texts, we measure the reconstruction log-likelihood of the test documents conditioning on the tags.

Specifically, for each test case $x_k$, which is a segment randomly sampled from a test document $x_{1...L}$ (randomly sampled from the test corpus), we measure the reconstruction log-likelihood of $x_k$ given latent tags $t_k$ under the LLM:

$$\mathbb{E}_{t_k \sim p_{LLM}(t|x_{1...k-1},x_k)}[\log p_{LLM}(x_k|x_{1...k-1}, t_k)] \tag{5}$$

Specifically, we first sample $S$ alternative segments at position $k$ independently by $\{\hat{x}_k^{(1)}, \hat{x}_k^{(2)}, ..., \hat{x}_k^{(S)}\} \sim p_{LLM}(\cdot|x_{1...k-1})$. Next, we conduct $T$ repeated experiments to approximate the log-likelihood of $x_k$ given the previous segments $x_{1...k-1}$ under the LLM. Each time, we randomly sample $C$ alternative segments from $\{\tilde{x}_k^{(1)}, \tilde{x}_k^{(2)}, ..., \tilde{x}_k^{(S)}\}$ and put it together with $x_k$ (in randomly shuffled order) as options and ask the LLM which one is the true continuation conditioned on $x_{1...k-1}$ and the tag $t_k$ predicted on $x_k$. Based on the number of times (denoted as $c_k$) that the LLM chooses $x_k$ as the true continuation among all $T$ experiments, we estimate the reconstruction log-likelihood with alpha-smoothing ($\alpha = 0.1$):

$$\mathbb{E}_{t_k \sim p_{LLM}(t|x_{1...k-1},x_k)}[\log p_{LLM}(x_k|x_{1...k-1}, t_k)] = \log \frac{c_k + \alpha}{T + \alpha S} \tag{6}$$

As a baseline, we compare the reconstruction log-likelihood with the log-likelihood without conditioning on any tags:

$$\mathbb{E}[\log p_{LLM}(x_k|x_{1...k-1})] \tag{7}$$

which we estimate in the same way as the reconstruction log-likelihood except that when asking the LLM to choose the true continuation, we only provide the previous text segments $x_{1...k-1}$ without any tags.

In our experiments, we estimate the log-likelihood on the same set of 1K randomly sampled test cases using each sampling method.

**Hits@K Accuracy** The latent tags can also be used for hierarchical generation where we first sample a sequence of tags as an outline and then sample the actual text based on the outline. To

evaluate if the latent tags help generate more diverse texts, we evaluate if the outputs cover more solution paths and thus lead to higher chance of hitting the correct path on problem solving tasks.

To this end, we evaluate the Hits@K accuracy of hierarchical sampling with latent tags, and compare it with the Hits@K accuracy of direct sampling without tags. Specifically, for each problem, we sample $K = 50$ solutions independently from an LLM given the problem description either directly or through hierarchical sampling with latent tags. If any of the $K$ solutions lead to the correct answer, it gets a score of 1, otherwise 0. Finally, we compute the average score over all testing problems.

For hierarchical sampling, we first sample a sequence of tags $(t_1, t_2, ..., t_l)$ (up till the special tag <END>) with maximum length L using a bigram model learned on the training data (based on the tag assignments):

$$p(t_1, t_2, ..., t_l) = p(t_1)p(t_2|t_1)...p(t_l|t_{l-1})p(\text{<END>}|t_l) \tag{8}$$

And then, we prompt the LLM to sample a solution to the given problem based on the sampled sequence of tags $(t_1, t_2, ..., t_l)$.

### 4.3 *fPLSA* SETUP

For the EM procedure, we set the maximum number of iterations to 30. At the E-step (where the LLM assigns a tag to each segment conditioned not only on the current segment but also on neighbouring segments within the context window), we use a context window size of 2 on WritingPrompts and use unlimited context window (such that the whole solution is used as context) on MATH and BBH. At the M-step, we randomly sample 10 segments assigned to each tag to update the tag description.

### 4.4 BASELINES

**TradLDA**  We compare our approach with the traditional Latent Dirichlet Allocation (*TradLDA*) algorithm designed to discover latent topics in a collection of text spans (Blei et al., 2003).

**TradLDA+LLM**  As Li et al. (2023) showed that the topic labels generated by LLMs based on the key terms learned through TradLDA are preferred more often than the original labels, we also include *TradLDA+LLM* as a baseline. Specifically, we first learn the topics with the key terms for each topic using TradLDA, and then use GPT-4 to generate a description for each topic based on the key terms.

**Prompting**  Recent work showed that, with appropriate prompts, LLMs are capable of directly generating topic labels given a set of text documents and condensing overarching topics (Mu et al., 2024). As a baseline, we adapt the approach (along with the prompts) to generate topic descriptions for each text segment.

**GenOutline**  For Hits@K accuracy, we also include a two-step sampling baseline, where we first prompt the LLM to generate a multi-step outline for solving this type of problem and then prompt the LLM to generate the actual solution based on the problem description and the outline.

### 4.5 LARGE LANGUAGE MODEL SETUP

For clustering and tagging, we use GPT-4 for all approaches, a powerful LLM (OpenAI et al., 2024). We set $top\_p = 0.5$, sampling temperature $\tau = 1.0$, zero frequency and presence penalty. We also use GPT-4 with $top\_p = 0.5$ to estimate the reconstruction log-likelihood. We set the temperature $\tau = 1.0$ when sampling alternative segments and $\tau = 0$ when choosing the best continuation.

To measure Hits@K Accuracy, we use ChatGPT (gpt-3.5-turbo; OpenAI (2023)) instead of GPT-4, because GPT-4 may have data contamination issues (Deng et al., 2024) on MATH and BBH benchmarks based on its timestamp. We set $top\_p = 0.5$ and temperature $\tau = 1.0$ when sampling solutions from ChatGPT.

|  | No Tag | TradLDA | TradLDA+LLM | Prompting | *fPLSA* |
|---|---|---|---|---|---|
| WritingPrompts | -4.81 | -3.75 | -4.12 | -3.62 | **-3.43** |
| MATH-Num | -3.32 | -2.96 | -3.28 | -3.06 | **-2.64** |
| MATH-All | -3.67 | -3.16 | -3.57 | -3.44 | **-3.04** |

Table 1: Reconstruction log-likelihood of *fPLSA* versus the baseline without tags (*No Tag*), traditional LDA (*TradLDA*), traditional LDA with LLM-generated tag descriptions (*TradLDA+LLM*) (Li et al., 2023), and the prompting baseline (*Prompting*) (Mu et al., 2024) on *WritingPrompts* story dataset, Number Theory dataset from MATH (*MATH-Num*), and the whole MATH (*MATH-All*) dataset.

| Keywords | Tag Description |
|---|---|
| nothing, get, life, else, light, across, best, ca, single, come, got, death, together, running, power, system, entire, could, control, everything | The words you've provided span a broad range of concepts, but they share a common denominator in that they can all be associated with themes commonly found in science fiction literature and media. |
| continued, surface, wait, raised, floor, slowly, give, new, sure, needed, around, also, face, body, fact, made, bitch, girl, guy, much | The words listed seem to be common English words that could appear in a wide range of contexts. However, given their generic nature, they could be particularly prevalent in narrative or descriptive writing, such as in fiction, storytelling, or personal narratives. |

Table 2: Examples of keywords learned on short story segments in WritingPrompts through *TradLDA* and the corresponding tag descriptions generated by GPT-4. Given only the keywords without context, the tag descriptions produced by GPT-4 are too generic to recover the original text spans.

| Prompting Tags | *fPLSA* Tags |
|---|---|
| Tag 1: Stories involving themes of sacrifice, duty, friendship, companionship, hope, and resilience in the face of crisis. | Tag 1: Scenes involving intense, often dangerous situations, like explosions, retreats, long nights, empty streets, fires, and storms. |
| Tag 2: Stories involving time travel, genetic irregularities, and strange creatures that feed on negative emotions. | Tag 2: The protagonist experiences surreal and unexpected events, often involving time travel or strange bodily functions, and narrates them in a casual, humorous tone. |
| Tag 3: Stories involving emotional moments and first hugs. | Tag 3: This tag is associated with story segments that feature intense emotional moments, often involving fear, anger, or distress, and frequently serve as turning points or climactic scenes in the narrative. |

Table 3: Example tags learned on short story segments in WritingPrompts through Prompting versus *fPLSA*. Prompting tags are either too mixed (e.g. Tag 1 and 2) or too generic (e.g. Tag 3), while *fPLSA* groups segments of similar themes into the same cluster and describes each cluster with detailed explanations and example plots.

## 5 RESULTS

### 5.1 RECONSTRUCTION LIKELIHOOD

First, we compare the reconstruction log-likelihood of *fPLSA* with the *No Tag* baseline (without conditioning on any tags). As shown in Table 1, conditioning on *fPLSA* tags helps predict the original texts: *fPLSA* brings 0.6–1.4 higher log-likelihood than the *No Tag* baseline.

*TradLDA* also brings higher reconstruction log-likelihood over the *No Tag* baseline. However, since *TradLDA* only captures word or term co-occurrences, it still underperforms *fPLSA* consistently on all three datasets. Moreover, *TradLDA+LLM* fails to improve over *TradLDA*. As shown by the examples in Table 2, it is extremely challenging for LLMs and even humans to extract meaningful semantic information from the keywords learned on short text segments through *TradLDA*, and the resulting tag descriptions are overly generic, making it challenging to reconstruct the original text segments accurately.

Compared with the Prompting baseline, *fPLSA* achieves 0.2–0.4 higher log-likelihood on all three datasets. We further compared the tags learned using Prompting versus *fPLSA*. As shown by the examples in Table 3, Prompting tends to merge unrelated topics into a mixed topic (e.g. Tag 1 and 2), and the resulting topics become overly broad. Even for tags sharing a common theme, the descriptions often lack specificity and detail (e.g. Tag 3). By contrast, *fPLSA* identifies segments with similar themes, groups them into a single cluster and produces more detailed tag descriptions with example plots.

## 5.2 HITS@K ACCURACY

|  | No Tag | GenOutline | TradLDA | TradLDA+LLM | Prompting | *fPLSA* |
|---|---|---|---|---|---|---|
| **MATH** | | | | | | |
| Algebra | 88.6 | 90.1 | **93.6** | 89.6 | 91.1 | 92.6 |
| Counting | 61.3 | 60.4 | 69.8 | 65.1 | 69.8 | **72.6** |
| Geometry | 53.1 | 55.2 | 58.3 | 57.3 | **62.5** | 60.4 |
| InterAlgebra | 55.7 | 51.7 | 58.7 | 59.2 | 61.2 | **64.7** |
| Number | 65.4 | 76.0 | 77.9 | 74.0 | **78.8** | **78.8** |
| PreAlgebra | 74.2 | 79.1 | 81.3 | 81.3 | **84.6** | 83.0 |
| PreCalculus | 42.2 | 46.8 | 51.4 | 46.8 | 49.5 | **54.1** |
| **Average** | 62.9 | 65.6 | 70.1 | 67.6 | 71.1 | **72.3** |
| **BBH** | | | | | | |
| Date | 92.8 | 94.4 | 95.6 | 95.2 | 95.2 | **98.8** |
| Formal | 45.2 | 61.2 | 65.6 | 52.8 | 57.2 | **93.2** |
| Geometric | 70.8 | 76.8 | 83.6 | 84.0 | 80.0 | **87.6** |
| Logical | 89.2 | 95.6 | 95.6 | 96.0 | 96.5 | **99.5** |
| Movie | 84.8 | 88.0 | 92.8 | 92.0 | 93.2 | **95.2** |
| ObjCount | 93.2 | 96.8 | 99.2 | **100.0** | **100.0** | 95.2 |
| Penguins | 93.8 | 99.3 | 99.3 | **100.0** | 99.3 | 99.3 |
| ReasonColored | 92.8 | 97.6 | 98.4 | 98.8 | 98.8 | **100.0** |
| RuinNames | 64.8 | 74.8 | 69.6 | 70.0 | 80.0 | **93.6** |
| TranslationError | 52.4 | 68.4 | 60.4 | 60.0 | 63.6 | **75.2** |
| Temporal | 86.4 | 98.4 | 93.2 | 96.8 | 98.0 | **100.0** |
| WordSort | 27.2 | 36.4 | 16.0 | 14.8 | 42.0 | **56.0** |
| **Average** | 74.5 | 82.3 | 80.8 | 80.0 | 83.7 | **91.1** |

Table 4: Hits@K accuracy of *fPLSA* versus directly sampling without tags (*No Tag*), two-step sampling with LLM-generated outline (*GenOutline*), traditional LDA (*TradLDA*), traditional LDA with LLM-generated tag descriptions (*TradLDA+LLM*) (Li et al., 2023), and the prompting baseline (*Prompting*) (Mu et al., 2024) on 12 challenging tasks from BBH benchmark (Suzgun et al., 2022) and 7 tasks from MATH (Hendrycks et al., 2021).

We further evaluate how the semantic structural tags help with downstream generation by measuring the Hits@K Accuracy of various sampling methods with or without tags. First, compared with direct sampling without using any tags, hierarchical sampling with *fPLSA* tags leads to significantly higher Hits@K accuracy by +9.4 points on MATH and +16.6 points on BBH on average. Additionally, we compare *fPLSA* with GenOutline, a two-step sampling approach where we prompt the LLM to generate an outline before generating the actual solution. GenOutline improves over direct sampling on most tasks, but still underperforms hierarchical sampling with *fPLSA* by 7–9 points. These results indicate that hierarchical sampling using tags derived from the domain-specific documents via *fPLSA*

**TradLDA Tags**

This cluster often contains words such as either, distinct, case, problem, must, find, 10, three, 72, follows, 3a, yields, since, digit, thus, digits, equal, 2a, 144, base.
This cluster often contains words such as 250, shown asy, makes, means, becomes, coordinates, sphere, origin, thus, left frac pi, frac pi right, pi right, cos frac, pi frac pi, pi pi, frac pi frac, pi frac, frac, frac pi, pi.
This cluster often contains words such as equation, note, sqrt, also, line, get, 2b, rightanglemark, abc, draw rightanglemark, 25 boxed, dfrac, must, since, let, expanding, property, 300, angle, xy.
This cluster often contains words such as 2t, makes, circ boxed, triangle, 120, 120 circ, 60 circ, 90 circ, circ angle, operatorname, 360 circ, 360, since, 45 circ, 180 circ, 90, 180, angle, 45, circ.
This cluster often contains words such as 40, also, overline, 14, therefore, bc, align therefore, end align therefore, circ, let, frac cdot, respectively, sqrt, triangle, cosines, law cosines, cdot, law, frac, angle.

**Prompting Tags**

Algebra and equations in manipulation and solving.
Algebraic manipulation and polynomial factorization.
Equation setup and solving for ages, distances, and quantities.
Inverse function calculations and summation.
Geometry and trigonometry in problem-solving.

***fPLSA* Tags**

Using congruence or similarity to deduce equal angles or sides in geometric figures.
Perform algebraic manipulations to solve for an unknown variable.
Utilizes specific mathematical theorems or properties, such as De Moivre's Theorem or the Law of Cosines, to solve problems.
Identify or prove relationships between angles, sides, or other elements of geometric figures.
This tag includes steps that conclude a mathematical procedure or finalize the simplification of an expression.

Table 5: Top 5 tags from TradLDA, Prompting and *fPLSA* that lead to the highest Hits@K Accuracy on MATH.

produces more effective and diverse output solutions, thereby increasing the likelihood of hitting the correct answer.

Next, we compare *fPLSA* with hierarchical sampling with existing tagging approaches. *fPLSA* tags lead to more diverse outputs with a higher chance of hitting the correct solution paths than TradLDA on 16 out of 19 tasks. It brings an an average accuracy improvement of 2–10 points over TradLDA. Similarly, compared with TradLDA+LLM, *fPLSA* achieves higher Hits@K Accuracy on 17 out of 19 tasks and improves the average accuracy by 5–11 points across BBH and MATH. Compared with the Prompting baseline, *fPLSA* achieves higher Hits@K Accuracy on 14 out of 19 tasks. Overall, hierarchical sampling with *fPLSA* tags improves Hits@K Accuracy over existing tagging approaches by 1–11 points on average.

We further examine the top 5 tags from each tagging approach that lead to the highest Hits@K Accuracy when used as part of the outline. As shown in Table 5, the TradLDA tags are too low-level, making it difficult for an LLM to follow. The Prompting tags, however, are too generic – for example, the tag "Algebra and equations in manipulation and solving" covers almost all solution steps in algebra problems. By contrast, *fPLSA* tags are more specific and instructive than the Prompting tags, but are still representative of groups of solution steps.

Finally, we investigate whether the tags learned through *fPLSA* generalize across tasks. Specifically, we examine the average Hits@K Accuracy of tags learned mostly from a particular task when used on other tasks. As shown in Figure 1, tags learned from tasks other than the test task are helpful in sampling effective solutions and sometimes even more helpful than the tags learned on the test task itself. This is possibly because the LLM is already familiar with the solution paths suggested by the tags learned from the test task itself, while the tags learned from other tasks may cause the LLM to think out of the box.

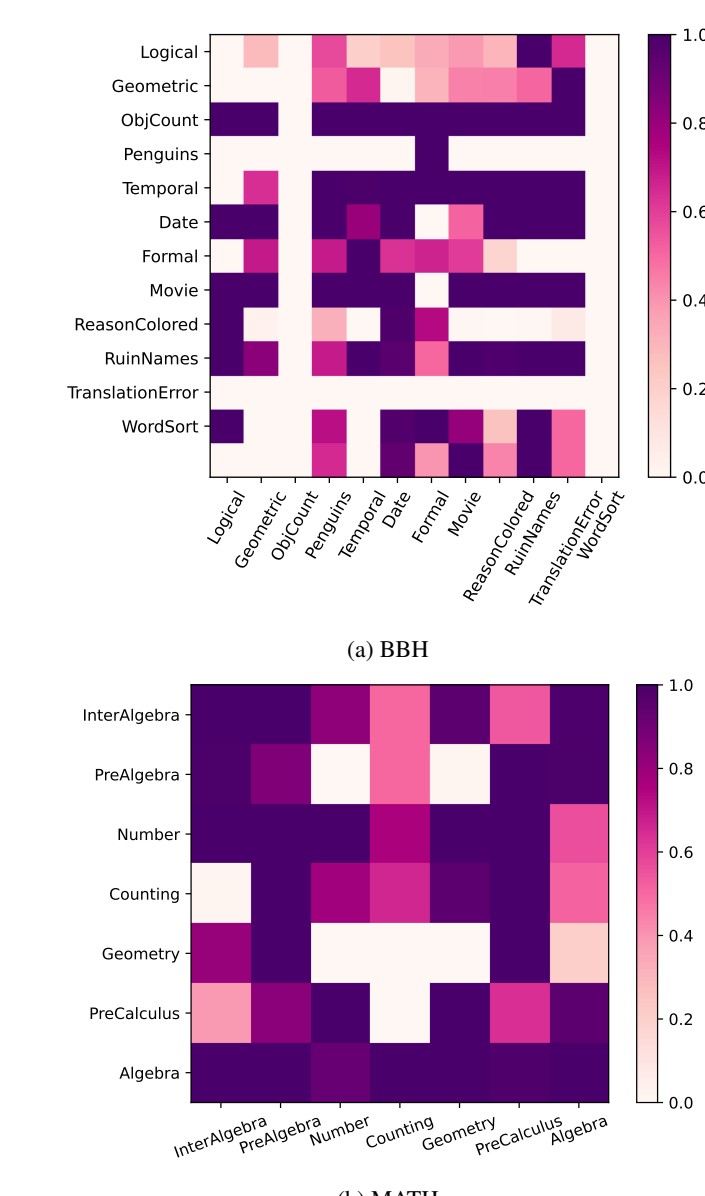

(a) BBH

(b) MATH

Figure 1: Heatmap of the average Hits@K Accuracy of tags learned mostly from a particular task when used on other tasks. The x axis represents the task from which the tags are learned from, and the y axis represents the test task. Tags learned from tasks other than the test task are proven to be helpful and sometimes even more helpful than the tags from the test task.

# 6 CONCLUSION

We introduced *fPLSA*, a foundation-model-based Probabilistic Latent Semantic Analysis method that aims to uncover the latent semantic structures in document collections by iteratively clustering and tagging document segments based on document-level contexts. Our experiments on story writing, math and multi-step reasoning tasks show that *fPLSA* tags are more informative in reconstructing the original texts than tags generated by existing tagging methods. *fPLSA* tags are also useful in generating more diverse solutions via hierarchical sampling and lead to higher Hits@K Accuracy than existing methods. These results suggest the potential of *fPLSA* for generating effective task guidelines given some worked-out examples, along with hierarchical sampling and searching for problem solutions based on a verification or reward model.

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
