# OpenReview forum: "fPLSA: Learning Semantic Structures in Document Collections Using Foundation Models"
_ICLR.cc/2025/Conference — Submitted to ICLR 2025_

### Official Review · Reviewer_Jd27 · 2024-10-27

**Soundness:** 2
**Presentation:** 1
**Contribution:** 2
**Rating:** 3
**Confidence:** 4

**Summary:**

### [Updates 12/04/2024]
To anyone reading the reviews: The authors published all their responses in the final hours of the rebuttal period (midnight EST). When I tried to respond just now, I realized I can no longer make my response public, so I'm updating my official review here. To clarify, **it was the authors who did not engage during the rebuttal period, not the reviewers**.

### Original Content
The paper introduces fPLSA, a foundation-model-based extension of Probabilistic Latent Semantic Analysis (PLSA), aimed at discovering semantic structures within document collections through clustering and tagging of text segments. Unlike traditional topic modeling, which often relies on word co-occurrences, fPLSA leverages Large Language Models (LLMs) to understand segment-level semantics in a broader document context. It applies an Expectation-Maximization (EM) algorithm to iteratively refine segment tags, enhancing both text reconstruction accuracy and hierarchical sampling. Experimental results on datasets for story writing, math, and reasoning show that fPLSA significantly outperforms traditional and LLM-based tagging methods in text reconstruction and solution diversity. This makes it suitable for generating effective problem-solving guides and diverse text outputs across varied tasks.

**Strengths:**

The article is generally well-written except for the technical part, which I find somewhat confusing. According to the article, fPLSA's strengths lie in its enhanced semantic understanding, leveraging LLMs for capturing nuanced document structures beyond lexical co-occurrence. This approach yields more accurate text reconstruction and supports hierarchical sampling, producing diverse, high-quality outputs in applications like story generation and problem-solving. Its specific and detailed tagging outperforms generic LLM-based tags, enhancing content generation. Additionally, fPLSA’s unsupervised clustering reduces the need for labeled data, while its demonstrated adaptability across domains and improved Hits@K accuracy make it a versatile, efficient tool for semantic analysis and structured text generation.

**Weaknesses:**

- The usage of math symbols is sometimes confusing. Also not required, it is suggested that the authors follow the [default notation](https://github.com/ICLR/Master-Template/raw/master/iclr2025.zip) for a clearer presentation of the equations.

- The proposed method is not thoroughly explained. For example, the computation of some terms in (4) is missing, as well as its optimization algorithm, e.g., how to calculate $p(x_k|d)$. From my perspective, if one chooses to express the idea using math formulas, then every term should be clearly explained except for the cases where it is extremely obvious, which I think does not apply to (4).

- A figure or algorithm may better explain the proposed method.

- The authors use GPT-4 for clustering and tagging but GPT-3.5 for response generation and did not provide experimental results on other combinations. The performance of the proposed method therefore may not be universally applicable.

-  Potential data leakage issues (detailed in Questions).

Overall, I think the approach proposed by this article is rather straightforward and could be easily described with better clarity without introducing any formulae, perhaps except for the motivation part. In addition, it seems that this article may find a broader audience in the pure NLP community rather than a mixed community of different machine learning topics. Therefore I would recommend to submitting this manuscript to ACL ARR venue instead of machine learning conferences.

**Questions:**

- I'm a bit confused by the relation between w, x, and d. If $w_{1:n}=x_k\subset d$, how is $p(x_k|d)$ modeled in (4)? Why is it necessary to include both $x_k$ and $d$ as conditional terms?

- Context window: I'm not sure I understand how the segments are selected in this article. Is a segment a fixed-length sequence of tokens? Are there any overlaps between different segments? In Line 238, the authors mentioned "we use a context window size of 2 on WritingPrompts and use unlimited context window". How should the "unlimited context window" be interpreted?

- According to [1], the latent variable ($z$ in [1]) is supposed to be categorical. This article borrowed the same concept from [1] but I'm not sure whether this article follows the original setup. The authors did mention that they "set the number of tags to 100", but the example tags in Table 3 showed that the tags are natural language descriptions rather than categorical labels. I wonder how the tags are generated, and if calling it "latent" is still appropriate.

- In (5), $t_k$ is sampled condition on $x_k$, which is later used to estimate the probability of reconstructing $x_k$. Is this a typo? Doesn't this lead to data leakage and make the results of (5) unfairly high?

- For BBH, I'm not sure why it is necessary to "use the step-by-step solutions produced by their automatic Chain-of-Thought
prompt inference algorithm for clustering and tagging". Does it mean that a part of the (ground-truth) solutions is utilized as the prompt to the model for problem-solving? I think this is a huge data leakage issue and would greatly undermine the soundness of the evaluation of the proposed method.

- Since tag generation is a recursive process, what would the token consumption be for achieving the presented results? How about the baseline models?

[1] Hofmann, T. "Probabilistic latent semantic indexing." Proceedings of the 22nd annual international ACM SIGIR conference on Research and development in information retrieval. 1999.

---

> ### Author Response · Authors · 2024-12-04
>
> Thank you for the insightful feedback!
>
> Responses to the weaknesses:
>
> 1.	Algorithm description: We will revise the algorithm description part to better explain the algorithm.
> 2.	GPT-4 for clustering and tagging but GPT-3.5 for response generation: While we use GPT-4 in most experiments, we use ChatGPT to generate the actual problem solutions in the accuracy experiment. This is because we notice that the zero-shot accuracy scores of GPT-4 on MATH and BBH benchmarks are very high already (which is possibly due to the data contamination issue [1]) and leave little room for improvements by exploring more diverse solution paths. Thus, we chose to use ChatGPT to generate the actual problem solutions in this experiment, which doesn’t cause unfair comparison because we compared with other tagging baselines that also use GPT-4 for clustering and tagging and ChatGPT for generating solutions. So the comparison is still fair.
>
> Responses to the questions:
>
> 1.	p(xk|d) refers to the distribution from which we randomly sample a text segment from a document. Empirically, we just randomly sample a segment index k from a uniform distribution over 1 to n, where k indicates which segment is the current one. We will revise the paper to better explain it.
> 2.	Segment length and context window: In terms of segment length, we take each paragraph as a segment for stories and each sentence as a segment for problem solutions. There is no overlap between segments. In terms of context window, it refers to the neighboring text segments to the current segment that we provide as additional context for tagging and clustering. A context window size of two means that we provide two neighboring text segments as context. And unlimited context window means that we provide all the other text segments in the same document as context.
> 3.	Categorical tags: The tags in our algorithm are also categorical. At the tag assignment step, the LLM is given the text information and all tags and is asked to choose the most suitable tag for the current text segment. The main difference between our algorithm and traditional PLSA is that, in PLSA, each tag is represented by a probability distribution over documents, while in our algorithm, each tag is represented by a textual description of a cluster of documents.
> 4.	Reconstruction likelihood: In (5), we measure the reconstruction likelihood, i.e. how well the learned tags help reconstruct the original text xk. This is a common way to evaluate latent variable models like VAE in both NLP and Vision.
> 5.	There is no data leakage in the evaluation. We learn the tags from step-by-step solutions from the training set and measure the Hits@K accuracy on a separate test set in which the problems are unseen.
> 6.	Computational cost: The exact token consumption depends on the document lengths, but the number of LLM calls for our algorithm is roughly (number_of_tags * number_of_iterations * 2 + number_of_segments), while the number of LLM calls for the prompting baseline is roughly (number_of_segments + 1). So, if the total number of segments in the document collection is large, the additional computational cost of our algorithm would be relatively small compared to the overall cost.
>
> [1] Chunyuan Deng, Yilun Zhao, Xiangru Tang, Mark Gerstein, Arman Cohan. Investigating Data Contamination in Modern Benchmarks for Large Language Models. NAACL 2024.

---

### Official Review · Reviewer_BWXW · 2024-11-03

**Soundness:** 3
**Presentation:** 3
**Contribution:** 3
**Rating:** 5
**Confidence:** 2

**Summary:**

The paper introduces fPLSA (foundation-model-based Probabilistic Latent Semantic Analysis), a novel approach for identifying latent semantic structures in document collections by combining traditional PLSA with the contextual understanding of large language models (LLMs). fPLSA enhances probabilistic clustering and unsupervised topic modeling by assigning semantic "tags" to document segments through an iterative Expectation-Maximization (EM) process, where each tag captures both local meaning and broader document context. This structured tagging approach enables fPLSA to better capture complex segment relationships, making it valuable for hierarchical sampling, document analysis, and potentially other downstream tasks such as structured summarization. The paper demonstrates fPLSA’s effectiveness across diverse datasets—narrative (story writing), problem-solving (math), and multi-step reasoning—showing improvements in text reconstruction likelihood and Hits@K accuracy, underscoring its robustness and versatility.

**Strengths:**

- **Innovative Approach:** fPLSA is a well-conceived combination of probabilistic topic modeling and LLM-based embedding, creating a tagging system that captures both low- and high-level semantics. This approach enables a nuanced understanding of document structure that extends beyond traditional methods, addressing complex relationships within text segments.
- **Diverse Evaluation:** The method is rigorously evaluated across multiple datasets, including narrative, mathematical, and multi-step reasoning tasks, demonstrating consistent performance improvements in text reconstruction and sampling diversity. This diversity in datasets reinforces the robustness and generalizability of the approach.
- **Potential for Cross-Domain Applications:** fPLSA’s ability to structure and tag text meaningfully is a powerful tool for hierarchical content generation, segmentation, and structured summarization, with substantial applications across various domains, such as education, content generation, information retrieval, and summarization.
- **Foundation for Future Research in Unsupervised Document Tagging:** fPLSA provides a strong foundation for future work in unsupervised document tagging and text segmentation. Its hierarchical tagging approach encourages further exploration in transfer learning, document summarization, and adaptive segmentation, inspiring new research directions for improved document understanding and organization.

**Weaknesses:**

- **Single-Document Applicability:** fPLSA heavily relies on cross-document patterns during training, which is not fully addressed in terms of single-document use cases. At test time, users often only have one document. It would be beneficial to clarify how fPLSA’s pre-trained tags would generalize to individual documents without access to cross-document patterns. For instance, can the model effectively apply pre-learned tags from similar training data to new documents?
- **Lack of Efficiency Analysis:** Given fPLSA’s reliance on LLMs, a discussion on computational efficiency would be valuable. While LLMs are powerful, they are computationally expensive. Addressing the practical feasibility of deploying fPLSA at scale (or proposing more efficient variations) would make the paper’s findings more actionable.
- **Potential LLM Biases:** Since fPLSA uses pre-trained LLMs to assign tags, there is a risk of encoding biases from the LLM's training data into the tags. The authors could explore ways to mitigate or assess the impact of these biases, especially for datasets or domains sensitive to fairness and accuracy.
- **Segmentation Granularity:** The paper does not discuss how sensitive fPLSA is to the choice of segment granularity (e.g., sentence, paragraph) and whether different segmentation approaches yield more cohesive or meaningful tags. Further examination of this could provide clarity on best practices for applying fPLSA across different document types and tasks.
- **Potential for Downstream Applications:** Although the paper’s results demonstrate fPLSA’s effectiveness in hierarchical sampling, the model's broader potential in downstream tasks is not explored. Given the rich, hierarchical nature of fPLSA tags, they could be valuable for applications like multi-level text summarization, where each tag could represent a theme or section for summarization. Exploring these applications would broaden fPLSA’s impact.

**Questions:**

- How would fPLSA perform when applied to a single document at test time, especially if that document differs significantly in structure or content from the training set? Can pre-learned tags from similar training data reliably generalize to new documents in such cases, or would fine-tuning on representative samples be necessary to improve performance?
- Given fPLSA’s structured tagging capabilities, could the authors discuss its applicability to downstream tasks like structured text summarization and content retrieval? Prior research on text summarization with text segmentation has demonstrated that segmenting texts by themes can enhance summarization quality [1]. Could fPLSA’s tags be similarly used to segment and summarize each thematic section, creating a coherent multi-level summary? Additionally, might these tags support content retrieval or indexing by allowing documents to be searchable by thematic segments? Including a brief paragraph on such applications could highlight the contribution's versatility.
- Can the authors provide insights into fPLSA’s computational cost compared to the baselines? For instance, would a less resource-intensive model (like a smaller language model) yield competitive results without the same computational burden?
- How sensitive is fPLSA to the choice of segment granularity (sentence, paragraph, etc.)? In testing, did certain segmentation approaches yield more cohesive or meaningful tags, and if so, could the authors elaborate?
- Since pre-trained LLMs may encode biases, did the authors observe any potential bias issues during fPLSA’s tagging process? If so, what mitigation strategies might they recommend for fair and balanced tag generation?

**References**
1. Semantic Self-Segmentation for Abstractive Summarization of Long Documents in Low-Resource Regimes. AAAI 2022.

---

> ### Author Response · Authors · 2024-12-04
>
> Thank you for the insightful feedback!
>
> Responses to the questions:
>
> 1.	How would fPLSA perform when applied to a single document at test time? Tags learned from the training documents through fPLSA can be directly applied to single document at test time, given that the test document shares similar characteristics with the training documents (e.g. if they are all math solutions or political speech documents).
> 2.	Applicability to downstream tasks like structured text summarization and content retrieval: Great point. Our algorithm can indeed be applied to improve text summarization and content retrieval as suggested.
> 3.	Computational cost: The exact token consumption depends on the document lengths, but the number of LLM calls for our algorithm is roughly (number_of_tags * number_of_iterations * 2 + number_of_segments), while the number of LLM calls for the prompting baseline is roughly (number_of_segments + 1). So if the total number of segments in the document collection is large, the additional computational cost of our algorithm would be relatively small compared to the overall cost.
> 4.	Choice of segment granularity: fPLSA can yield meaningful segment tags given that each segment contains meaningful information. For instance, on the story dataset, we segment the stories by paragraphs, while on the math solution dataset, we segment the solutions by sentences.
> 5.	Potential biases from LLMs: LLM may introduce model bias in the learned tags, although we didn’t observe any on our current evaluation datasets. We can potentially apply existing bias mitigation methods in our algorithm, which we leave for future work.

---

### Official Review · Reviewer_6feW · 2024-11-04

**Soundness:** 2
**Presentation:** 2
**Contribution:** 2
**Rating:** 3
**Confidence:** 3

**Summary:**

This paper introduces an improved version of probabilistic Latent Semantic Analysis (pLSA), termed fPLSA (Foundation-Model-Based PLSA), which incorporates Large Language Models (LLMs) to refine the modeling of latent tags in documents for topic modeling. It conducts some experiments to verify the effectiveness of fPLSA.

**Strengths:**

(1) Study a classic task

(2) Propose a new method

(3) conduct some experiments to verify the effectiveness of the proposed method.

**Weaknesses:**

(1) Insufficient technical contribution: The method utilizes LLMs for tag Description Generation. Specifically, the fPLSA model  generates descriptive tags for the document segments by prompting the LLM with segments assigned to a particular tag to produce a cohesive summary that represents what these segments have in common. The parameters of the LLM are kept frozen during the process. This means the LLM is not fine-tuned during fPLSA training but is used in a static manner. While the integration of LLMs into pLSA offers a novel approach to document modeling, the core statistical methodologies underlying pLSA (like the EM algorithm) remain largely unchanged. This may limit the perceived novelty from a methodological standpoint.

(2) Missing necessary experiments: need to involving more baselines that use LLMs for topic modeling, like Pham et al. (2024), Wang et al. (2023) mentioned in the paper.

(3) Poor writing: The transition of some contents are abrupt and hard to readers to understand the points, such as the first and second  paragraphs in the introduction.

(4) Missing Implementation Details: all the prompts used in the experiment are not specified such as those for fPLSA and GenOutline (a baseline)

(5) Unclear motivation of the experiment setting: the paper uses GPT-4 for clustering and tagging  while using ChatGPT to measure the accuracy. The authors explain it’s because GPT-4 may have data contamination issues on some benchmarks. I think this explanation is lame and need more clarifications while potentially leading to unfair comparison.

**Questions:**

please refer to the weaknesses

---

> ### Author Response · Authors · 2024-12-04
>
> Thank you for the insightful feedback!
>
> Responses to the weaknesses:
>
> 1.	Insufficient technical contribution: This is the first work (as far as we know) that combines the merits of LLM and traditional topic modeling algorithms, and has shown outstanding improvements by combining these two types of approaches.
> 2.	Missing necessary experiments: Pham et al. (2024) and Wang et al. (2023) are both prompt-based topic modeling approaches. We have included a representative prompting method (Mu et al. 2024) in our experiments.
> 3.	Writing: We will revise the paper to address the issue.
> 4.	Missing Implementation Details: We will update the appendix to include the prompt templates.
> 5.	Motivation for using ChatGPT to measure accuracy: While we use GPT-4 in most experiments, we use ChatGPT to generate the actual problem solutions in the accuracy experiment. This is because we notice that the zero-shot accuracy scores of GPT-4 on MATH and BBH benchmarks are very high already (which is possibly due to the data contamination issue [1]) and leave little room for improvements by exploring more diverse solution paths. Thus, we chose to use ChatGPT to generate the actual problem solutions in this experiment, which doesn’t cause unfair comparison because we compared with other tagging baselines that also use GPT-4 for clustering and tagging and ChatGPT for generating solutions. So the comparison is still fair.
>
> [1] Chunyuan Deng, Yilun Zhao, Xiangru Tang, Mark Gerstein, Arman Cohan. Investigating Data Contamination in Modern Benchmarks for Large Language Models. NAACL 2024.

---

### Official Review · Reviewer_pfT6 · 2024-11-05

**Soundness:** 2
**Presentation:** 2
**Contribution:** 2
**Rating:** 3
**Confidence:** 4

**Summary:**

This paper studies the problem of discovering text segments that share common characteristics and assigning them the same tag description. It proposes an LLM-based method that iteratively assigns tags to document segments and improves each tag description based on the segment cluster. The authors aim to show that these tags are helpful for a reconstruction task and in improving “Hits@K” accuracy in evaluation sets created from WritingPrompts, MATH, and the BBH benchmark.

**Strengths:**

- This work proposes to discover “tags” for text segments in an unsupervised fashion and a novel algorithm that is inspired by the probabilistic latent semantic analysis (PLSA).
- The algorithm leverages the ability of an LLM to analyze textual materials and is able to find detailed and meaningful tags, as shown in the qualitative results of the paper.
- The paper show favorable empirical results compared to multiple baselines: traditional latent Dirichlet allocation, it variant + LLM, prompting, and chain-of-thought prompting.

**Weaknesses:**

1. The writing in this paper is often too generic and high-level. For example, when a reader reads the motivation at L011-L014 “Humans have the ability to learn new tasks by inferring high-level concepts from existing solutions, then manipulating these concepts in lieu of the raw data. Can we automate this process by deriving latent semantic structures in a document collection using foundation models?”, they may wonder:
    - What tasks do you mean?
    - Existing solutions of the “new tasks” or other relevant tasks?
    - What does it mean to manipulate high-level concepts?
    - How do you define “semantic structures” in a document collection? It’s not precise to describe a set of “tags” as a structure.
2. The novelty of the method is limited and its connection to PLSA and EM is loose. The proposed algorithm is simple: (1) Initialize a certain number of tag descriptions. (2) Prompt an LLM to assign a tag to each document segmentation based on the tag descriptions. (3) Let an LLM generate a new tag description that describes the shared characteristics of the segments in this cluster.
    - The main Eq. (4) is actually not used: $p(d)$, $p(x_k|d)$,  $p_\Theta(t|x_k,d)$, and $p_\Theta(w_{1\dots n}|t)$ are not computed.
    - L153: The parameters $\theta_t$ are textual descriptions instead of floating-point parameters and no training is happening.
    - L157: No probability distribution is involved. An LLM is employed to greedily perform the steps in the algorithm.
    - PLSA is a generative model of the training documents that it is estimated on, and it is not a generative model of new documents. But this paper aims to find tags that apply to unseen examples.
3. While the convergence criteria matters for an EM algorithm, this paper simply sets the number of iteration to 30. Not enough analyses is perform on the impact of the number of iterations.
4. In the reconstruction experiments the method based on learned tags solves a multiple choice problem of picking the ground truth $x_k$ from a set of candidate segments. However, baselines such as prompting in Eq. (7) requires a language model to generate the ground truth $x_k$. These seem not comparable.
5. Although the experiment results are positive compared to the baselines, the setups are synthetic. Would be nice to see the application of this algorithm to achieve competitive results according to standard evaluation metrics of the used datasets, which are common benchmarks.
6. Many details are missing. See the Questions below.

**Questions:**

1. What prompt templates are used in various experiments?
2. How are the textual descriptions $\theta_t$ initialized in the algorithm? How can the initial tag descriptions meaningfully be assigned to the segments?
3. Sec 4.1 Evaluation Datasets: How do you convert the input query and the output answer of each example into segments? Do you learn tags only for segments of answers, but not queries/prompts/questions?
4. Sect 4.1: This is titled Evaluation Datasets but indeed describes data for clustering and tagging.
5. L195: How do you sample alternative segments?
6. L188: What do you mean by the test documents? The datasets are query-answer examples.
7. In the Hits@K evaluation:
(a) Do you first generate the tag sequence based on the input of a test example or randomly?
(b) Does a model predict an answer based on the tag sequence in one prompting call?
(c) How do you evaluate if a sampled solution is correct or not?
(d) Why do you say the proposed algorithm improves diversity in outputs, which is not evaluated? In fact, diversity is neither necessary nor sufficient for a model to perform a reasoning task well.

---

> ### Author Response · Authors · 2024-12-04
>
> Thank you for the insightful feedback!
>
> Responses to the weaknesses:
>
> 1.	Clarifying some misunderstandings of the algorithm:
>     -	The proposed algorithm is iterative, in which the tag “parameters” and assignments are iteratively updated, same as the EM algorithm, instead of the three-step procedure as described in the review.
>     -	Eq.(4) represents the generative model of the text in each document. Similar to the standard PLSA algorithm, p(d) and p(xk|d) represents the empirical distribution of the documents d and the segments  xk in each document, from which we sample the documents and document segments. pΘ(t|xk,d) and pΘ(w1…n|t) are optimized through our EM algorithm, which finds the Maximum A Posteriori estimates of Θ. We followed the notations and math formulas in the original PLSA paper in our paper.
>     -	L153: The parameters being updated in our algorithm are the textual descriptions of the tags, so there is still training. It’s just that the parameters being updated are discrete tokens.
>     -	L157: The tag assignment procedure is done by prompting the LLM with temperature=1, so it’s still a probabilistic sampling procedure based on the current tag descriptions. We will revise the paper to clarify that.
>     -	According to the original PLSA paper, “To derive conditions under which generalization on unseen data can be guaranteed is actually the fundamental problem of statistical learning theory.” In other words, the learned tags can generalize to unseen data under certain conditions based on statistical learning theory.
>
> 2.	We set the maximum number of iterations to 30 based on our observations in preliminary experiments that the learned tag descriptions become stable in less than 30 iterations.
> 3.	Clarification on the evaluation method: The evaluation method is the same for all baselines and our method. When evaluating both the baseline without tags, baselines using other tagging algorithms, and the tags learned using our algorithm, we prompt the model to solve a multiple-choice problem of picking the ground truth xk from a set of candidate segments.
> 4.	Please point us to any existing evaluation metrics for document segment tagging, because we could only find existing evaluation metrics for topic modeling of whole documents.
>
> Responses to questions:
>
> 1.	Prompt template: We will update the appendix with the prompt template.
> 2.	Initialization of the tag descriptions: Initially, we randomly assign some text segments to each tag and prompt the model to summarize their commonalities as the initial tag description.
> 3.	Sec 4.1 Evaluation Datasets: We learn tags from solution segments only.
> 4.	Sect 4.1: In Section titled “Evaluation Datasets”, we describe the datasets used for both learning the tags and evaluating the quality of these tags.
> 5.	L195: As described in L196, we sample the alternative segments from the LLM given the previous segments in the same document.
> 6.	L188: On the story dataset (WritingPrompts), the test documents refer to the stories themselves. On MATH, the test documents refer to solution texts.
> 7.	In the Hits@K evaluation: (a) The tag sequence is generated randomly without given the test example/question. (b) The model predicts the answer based on the tag sequence in one prompting call. (c) We measure if a sampled solution is correct or not by checking if the final answer in the solution is the correct answer. (d) We need diversity in the generated outputs because on challenging reasoning tasks, we may need to search through the output space to find a better answer, and the searching algorithms wouldn’t work if the generated outputs are all similar. To this end, we measure if sampling with our tags can help generate more diverse solution paths in the way that increases the chance of finding a correct solution among all sampled outputs.

---

### Meta-Review · Area_Chair_DPX1 · 2024-12-20

**Metareview:**

This paper proposes a new method to tag documents based on high-level concepts, using an LM-based Probabilistic Latent Semantic Analysis. The paper shows this can help reconstruct the original texts from tags, and can be used for hierarchical sampling with more diverse outputs.

Strengths
- The paper proposes a new way to tag text documents without any supervision (PLSA, 6feW, BWXW, Jd27).
- Empirical results are stronger than multiple baselines (PLSA, Jd27).

Weaknesses
- Writing needs improvements (PLSA, 6feW, Jd27).
- Insufficient technical contribution (PLSA, 6feW).
- Setups in the experiments are too synthetic (PLSA).
- Insufficient necessary experiments, such as comparison to prior LM-based topic modeling methods (6feW).
- Many missing details and ablations (PLSA, 6feW, BWXW, Jd27).

**Additional Comments On Reviewer Discussion:**

A few clarification was provided during rebuttal, but the changes needed to incorporate them to the paper is significant.

---

### Decision · Program_Chairs · 2025-01-22

Reject